# Surgical and Bioengineering Integration in the Anatomy Course of Medicine and Surgery High Technology: Knowledge and Perception of Anatomy

**Selenia Miglietta** [1], **Giuseppe Familiari** [1], **Michela Relucenti** [1,*], **Stefania Basili** [2], **Fabiano Bini** [3], **Gabriele Bove** [4], **Claudio Barbaranelli** [5] and **Pietro Familiari** [6]

1   Department of Anatomy, Histology, Forensic Medicine and Orthopedics, Sapienza University of Rome, 00161 Rome, Italy
2   Department of Translational and Precision Medicine, Sapienza University of Rome, 00185 Rome, Italy
3   Department of Mechanical and Aerospace Engineering, Sapienza University of Rome, 00184 Rome, Italy
4   I.N.I. Group, Orthopaedic Unit, Grottaferrata, 00046 Rome, Italy
5   Department of Psychology, Sapienza University of Rome, 00185 Rome, Italy
6   Division of Neurosurgery, Department of Human Neurosciences, Policlinico Umberto I, Sapienza University of Rome, 00185 Rome, Italy
*   Correspondence: michela.relucenti@uniroma1.it

**Abstract:** The Locomotor System Anatomy (LSA) course, placed in the first semester of the first year of the new Master's degree in Medicine and Surgery High Technology (MSHT) at the Sapienza University of Rome, was integrated with surgical and bioengineering content. This study investigated the educational value and the students' perceptions of the effectiveness of these two types of integration, comparing surgical integration (SI) with engineering integration (EI). Anatomy knowledge and students' opinions attending the LSA course in MSHT degree (n = 30) were compared with those of students (n = 32) attending another medical and surgery course not comprising EI. Data show that students in the MSHT course like in-depth SI much more than in-depth EI. However, those who like in-depth SI also like in-depth EI. Significant differences were in anatomy knowledge between the two groups in the three sections of the test. There was no significant correlation between the three test scores and the levels of liking, while there was a significant correlation between students liking SI and those liking EI. A statistically significant correlation was also found in students who correctly responded to questions on the head and trunk, with students responding correctly to questions on the upper limbs. This study will be important in optimizing the deepening of SI and EI in the LSA course.

**Keywords:** human anatomy; anatomical sciences education; gross anatomy teaching; locomotor system; neurosurgery; orthopedics; surgical integration; bioengineering integration; technical physician; technical medicine

## 1. Introduction

The extreme dynamism and continuous introduction of new technologies that improve patient care and preventive health care characterize modern medicine; thus, the implementation of the educational curriculum of medicine and surgery with bioengineering contents is fundamental. Such educational curriculum modernization is aimed at creating future physicians with skills for digital health and quality improvement, a mindset for precision and personalized medicine. Health professionals, informed about the advantages of artificial intelligence, machine learning, medical robotics, network medicine, big data analysis, genomics, omics, and all other technologies related to bioengineering, bioinformatics, and bioelectronics will be introduced in this work [1–4]. The first Master's degree course in Medicine and Surgery High Technology (HT) in the Italian State University system was activated at the Sapienza University of Rome in the academic year 2020–2021 [5,6]. Strong vertical integration of basic and clinical sciences characterizes this degree course [5,7];

thus, the Human Anatomy course (placed in the first and second years) is integrated with surgical and bioengineering content. In particular, the Human Anatomy 1 module (focused on the locomotor system) is articulated with anatomy lessons interspersed with surgical lessons (given by orthopedists and neurosurgeons, who collaborate in subsequent modules of the anatomy course [8]) and is further implemented with a module on biological systems mechanics and biomechanics of the locomotor system (this module is carried out by an industrial bioengineering teacher). While the integration of anatomy with the clinical sciences is now well established in anatomy teaching [9,10], the integration with a bioengineering discipline is a major pedagogical innovation. Therefore, in this study, we investigated the educational value and the students' perceptions of the effectiveness and usefulness of this new type of integration compared to surgical integration.

## 2. Materials and Methods

### 2.1. The Undergraduate Course of Medicine and Surgery HT at Rome Sapienza University's Medical School

The undergraduate course is held at the Policlinico Umberto I Hospital associated with the faculty; the biomedical-technological training program consists of a 6-year curriculum, designed in collaboration with the Faculty of Medicine and Engineering, and trains students in both technological and bioengineering skills [6].

This new undergraduate curriculum was activated in the academic year 2020–2021 and was divided into 12 semesters and included 36 integrated courses with related exams. Bioengineering sciences are added to the vertical and horizontal integration of basic and clinical sciences during the six years of the degree program. This curriculum organization reduces the emphasis on teacher-centered lectures and focuses on a more student-centered learning model. For this purpose, activities, including practical integrated experiences and tutorials or seminars, were introduced since the first-year course [5,6,8].

### 2.2. The Course of the Locomotor System of Medicine and Surgery HT at the Sapienza University of Rome

The locomotor system course is a part of the initial stage of the human anatomy curriculum taught during the first-year's first semester of the undergraduate course. It comprises face-to-face lessons as well as practicals when students in small groups use plastic models, histology slides, and interactive multimedia tools proposed by the teacher.

Clinically-integrated lectures were organized as a presentation of clinical cases by an orthopedic surgeon or a neurosurgeon in the presence of an anatomy teacher, who actively contextualizes the presentation of surgical cases.

Surgical integrated lessons consisted of the clinical case presentation: the orthopedic surgeon illustrated cases of the shoulder, hip, and knee joint surgery, emphasizing the technical aspects of the operative procedures, including robotic surgery (examples of the orthopedic surgeon's teaching activities are shown in Figure 1).

The neurosurgeon presented the cases of brain and spinal cord surgery, describing the particular operative techniques of intervention, such as the different types of craniotomies and the different surgical approaches to the spine. Special emphasis was given to illustrating the advantages of innovative neuronavigation techniques concerning traditional approaches (examples of neurosurgeon's activities are shown in Figure 2).

As a further activity in addition to the lectures, small groups of students were organized to train their technical and surgical skills by simulating both orthopedic robotic surgery (Mako robotics–Stryke®, Kalamazoo, MI, USA) and neurosurgery with the aid of 3D simulation and neuronavigation tools (Brain-Lab®, Munich, Germany).

The bioengineering-integrated lectures, given by a teacher in industrial bioengineering, were intended to provide fundamental knowledge about the study of motion and stresses in biological systems to forces caused by biomechanical phenomena. The basic principles of computational biomechanical analysis of a multi-link model of the human body were also explained. Frontal teaching was devoted to the reference system for biomechanical analysis, degrees of freedom of joints, elementary movements and kinematic models of

limbs, and geometry of masses. The hands-on exercises used specific platforms to illustrate the principles of numerical and computational methods for biomechanical analysis and were devoted to the modeling of the limb to analyze and simulate muscle actions and exchanged forces. Examples of the bioengineering teacher's teaching activities are shown in Figure 3.

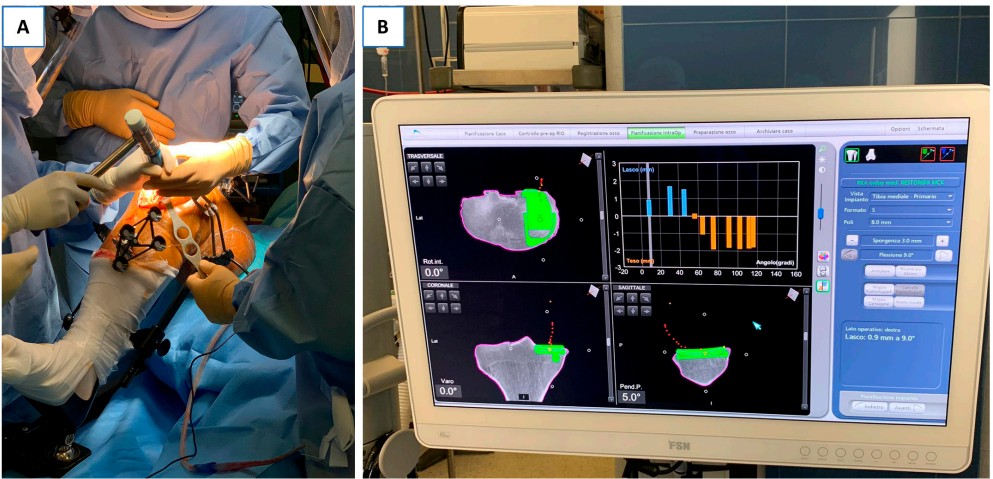

**Figure 1.** (**A**) Knee joint prosthetic implant surgery using the Mako robotics–Stryker® robotic system. Note the presence of the navigation system applied to the patient's leg. (**B**) Control monitor of Mako robotics–Stryker® robotic system. During knee joint implant surgery, the surgeon can observe the removed bone surface while performing the procedure and placing the prosthesis.

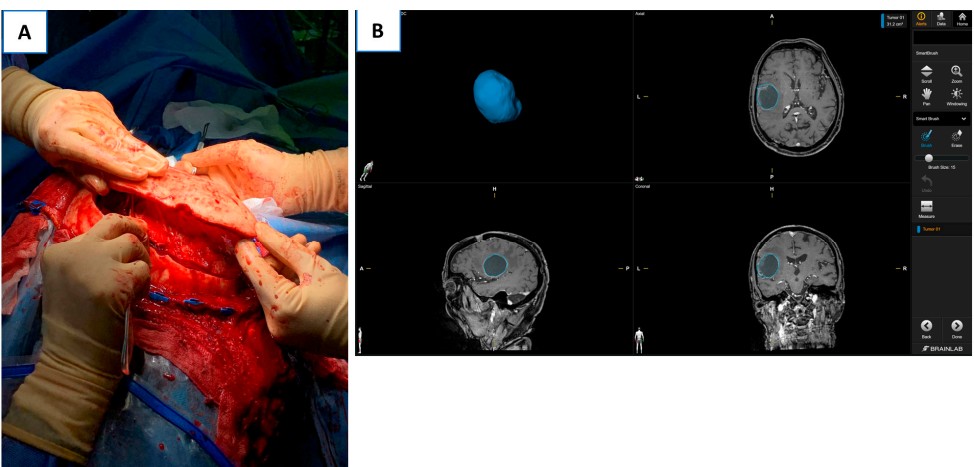

**Figure 2.** (**A**) Separation of the craniotomy bone operculum from the dura mater during decompressive craniotomy surgery for the evacuation of acute subdural hematoma. (**B**) BrainLab® neuronavigation monitor during neurooncological surgery. The 3D reconstruction of the intra-axial neoplastic lesion: axial, sagittal, and coronal of the tumor lesion inside the skull.

### 2.3. Student Sampling

Students (n = 62) who attended at least 67% of the mandatory locomotor system anatomy course during the academic year 2021–2022 were the subject of this study. The ages of the students ranged between 19 and 20; 65.6% of them were female, and 34.4% were male. Students (n = 30) from the first year of the course degree in medicine and surgery HT (Faculty of Medicine and Dentistry) were considered the study group, while students (n = 32) attending at least 67% of the mandatory anatomy course of the locomotor system and belonging from the first year of another degree in medicine and surgery course degree not comprising engineering integration (Faculty of Medicine and Psychology), acted as the control group. The anatomy and clinical teachers were the same in both the HT Medicine

course at the Faculty of Medicine and Dentistry and the Medicine and Surgery course at the Faculty of Medicine and Psychology.

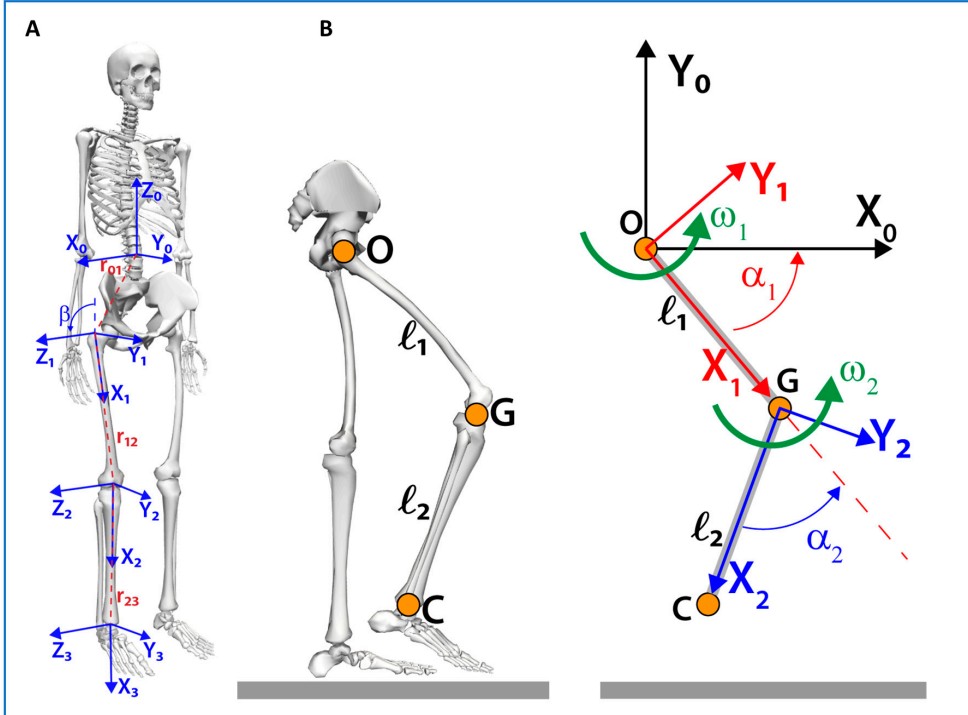

**Figure 3.** (**A**) Illustration of the coordinate systems of human skeletal structures: torso coordinate system X0Y0Z0 with origin coincident with the barycentre of the body, hip joint coordinate system $X_1Y_1Z_1$, knee joint coordinate system $X_2Y_2Z_2$, and ankle joint coordinate system $X_3Y_3Z_3$. Vectors $r_{01}$, $r_{12}$, and $r_{23}$ connect the coordinate system [i-1] with the system [i], where i = 1,2,3. The coordinate system (1) is obtained from the translation of the coordinate system(0)in the center of the hip joint and the subsequent rotation around the $Y_1$ axis by an angle β. (**B**) Kinematic analysis of the lower limb using a simplified model composed of two segments of length $\ell_1$, and $\ell_2$, respectively, knowing the system configuration through the angles α and the joint angular velocities ω.

### 2.4. Students' Views

The questionnaire used in this study was adapted from a questionnaire previously used in our study dealing with the integration of neurosurgery in neuroanatomy, where the Cronbach alpha was 0.9707 [8]. It was administered after the last lecture on the locomotor system course to all Medicine and Surgery HT students (n = 30) who attended the entire cycle of lessons. The students were asked to fill out the questionnaire anonymously. Informed consent to participate was obtained from each student after explaining the objective and purpose of the study.

The questionnaire was divided into two sections, A and B.

Section A of the questionnaire was designed to gather data concerning the didactic usefulness of clinically-integrated or engineering-integrated learning. Subsection A1 evaluated the usefulness or the uselessness of the surgical integration, whereas Subsection A2 evaluated the usefulness or the uselessness of the engineering integration.

Section B of the questionnaire was designed to collect data on the didactic usefulness of clinical case presentations or principles of computational biomechanical analysis modeling. Subsection B1 evaluated the usefulness or the uselessness of the lessons, including orthopedic and neurosurgical case presentations, while Subsection B2 evaluated the usefulness or the uselessness of the lessons in which modeling computational biomechanical analysis principles have been presented.

Each Subsection, A1/A2 and B1/B2, contained four topic-related items. Two of these items were positive, while the other two were negative, in agreement with a twofold

cross-check. A Likert scale was applicated to each item to test the agreement of the students. Respondents were invited to indicate their agreement on a five-point Likert scale (1 = strongly disagree, 2 = disagree, 3 = neither agree nor disagree, 4 = agree, 5 = strongly agree).

### 2.5. Data Analysis of Students' Views

Individual responses obtained using the Likert scale were treated as variables measured at the level of equivalent intervals, and then mean standard deviations were computed being an adequate statistic for this measurement level [11] (regarding treating Likert scales at the level of equivalent intervals, see also [12]). Pairwise comparisons of sections A1 vs. A2 and B1 vs. B2 were performed using the t-student test. Statistical significance was established at $p \leq 0.05$. The internal consistency of the data was assessed using Cronbach's Alpha calculations. A Cronbach's Alpha value higher than 0.70 is considered adequate for the internal consistency of the questionnaire [13]. Data were analyzed using IBM-SPSS 27.

### 2.6. Assessment of Students' Knowledge of Locomotor System Anatomy

The students' knowledge of the functional anatomy of the locomotor system was assessed using a test containing 30 questions on the functional anatomy of the locomotor system taken from a human anatomy text containing a collection of multiple-choice questions for testing and self-testing of learning used by Italian students [14]. The test was structured into 30 questions, out of which 11 concerned the functions of the muscles of the head and neck, another 11 pertained to the functions of the muscles of the upper limb, and, finally, eight questions were inherent to the actions of the muscles of the lower limb. The same test was administered, after the last lecture on the locomotor system anatomy course, to students belonging to the Medicine and Surgery HT degree (study group, n = 30) and to students belonging to another degree in medicine and surgery at Sapienza University of Rome (control group, n = 32). The 30 test questions are provided as material in Appendix A of this scientific article.

### 2.7. Data Analysis of the Assessment of Students' Knowledge of Locomotor System Anatomy

The sum of the correct answers in the three sections of the administered test questions on locomotor system anatomy was considered a measure of learning. Differences between traditional and HT students in these three scores were analyzed using the ANOVA test. Statistical significance was established at $p \leq 0.05$. Data were analyzed using IBM-SPSS 27.

### 2.8. Correlation Analysis between Student Opinions and Anatomy Knowledge Results

Correlation analysis among students' opinions on the usefulness of clinical integration or engineering integration and results obtained in the assessment test of students' functional anatomy knowledge was performed using the Bravais–Pearson coefficient. Statistical significance was established at $p \leq 0.05$. Data were analyzed using IBM-SPSS 27.

## 3. Results

### 3.1. Students' Opinions

#### 3.1.1. Evaluation of the Usefulness of Clinically-Integrated versus Engineering-Integrated Lessons in Learning Functional Anatomy

Data are presented in Table 1 in two distinct subsections: Subsection A1 contains data concerning the usefulness or the uselessness of surgical integration, whereas Subsection A2 presents data on the usefulness or the uselessness of the bioengineering integration.

**Table 1.** Student view of didactic usefulness of surgical vs. bioengineering integrated class usefulness in learning functional anatomy.

| Subsection A1 Surgical Integration: | Mean (±SD) [a] Min–Max | Subsection A2 Bioengineering Integration: | Mean (±SD) [a] Min–Max |
|---|---|---|---|
| It is useful to improve general knowledge of locomotor system functional anatomy (A1.1) | 4.33 (±0.884) 2–5 | It is useful to improve one's general knowledge of locomotor system functional anatomy (A2.1) | 3.70 (±0.750) 2–5 |
| Makes the lessons dynamic and interesting (A1.2) | 4.50 (±0.572) 3–5 | Makes the lessons dynamic and interesting (A2.2) | 3.20 (±0.925) 1–5 |
| It is totally useless for improving one's general knowledge of the locomotor system functional anatomy (A1.3) | 1.70 (±1.022) 1–5 | It is completely useless for improving general knowledge of locomotor system functional anatomy (A2.3) | 2.07 (±0.785) 1–4 |
| Makes lessons hard and boring (A1.4) | 1.43 (±0.568) 1–3 | Makes lessons hard and boring (A2.4) | 2.60 (±0.894) 1–4 |

[a] Likert scale: 1 = strongly disagree, 2 = disagree, 3 = neither agree nor disagree, 4 = agree, 5 = strongly agree; Cronbach's alpha = 0.876.

As shown in Table 1, the answers to statement A1.1 (opposite A1.3) showed that a very high number of students stated that clinical integration of the locomotor system was useful for improving one's general knowledge of locomotor system functional anatomy. Even responses to statement A2.1 (opposite A2.3) showed that a high number of students stated that the engineering integration of the lessons from the locomotor system lessons was useful for improving general knowledge of locomotor system functional anatomy.

In Table 2, the results of the paired *t*-test with comparisons between statements of Section A1 and Section A2 are presented. Data revealed that student satisfaction was higher for surgical integration compared to satisfaction for engineering integration.

**Table 2.** Paired *t*-test results with comparisons between statements of Sections A1 and A2.

| Statements | Mean Difference | T |
|---|---|---|
| A1.1 vs. A2.1 positive | 0.633 | 3471 ** |
| A1.2 vs. A2.2 positive | 1300 | 7208 *** |
| A1.3 vs. A2.3 negative | −0.367 | −1690 |
| A1.4 vs. A2.4 negative | −1167 | −7000 *** |

The degrees of freedom are 29 for all tests. ** $p \leq 0.01$, *** $p \leq 0.001$.

Furthermore, the answers to statement A1.2 (opposite to A1.4) revealed that a very high number of students stated that the surgical integration of the locomotor system makes the lessons dynamic and interesting. The answers to Statement A2.2 also reflected students' positive assessment of engineering integration, although this evaluation was significantly lower than the evaluation of surgical integration evaluation.

However, the responses to the questions about the perceived lack of usefulness or boringness of these lessons show very low values, although, in this case, there is a statistical difference in the difficulty of engineering integration, which prevails over the difficulty found in surgical integration. In contrast, there were no statistically significant differences in perceived low usefulness for surgical or engineering integration, both of which show very low values.

3.1.2. Evaluation of Clinically-Integrated or Engineering-Integrated Lessons' Usefulness in Learning Morphological Anatomy

Data are presented in Table 3 in two distinct subsections: Subsection B1 contains data concerning the usefulness or the uselessness of lessons, including surgical cases, whereas Subsection B2 contains data on the usefulness or the uselessness of the lessons, including modeling computational biomechanical analysis principles.

**Table 3.** Students' views of the didactic usefulness of surgical case presentations or principles of modeling the computational biomechanical analysis.

| Subsection B1 Lessons Including Surgical Cases: | Mean (±SD) [a] Min–Max | Subsection B2 Lessons Including Modeling Computational Biomechanical Analysis Principles: | Mean (±SD) [a] Min–Max |
|---|---|---|---|
| B1.1 improve one's general knowledge of locomotor system morphodynamics | 4.30 (±0.794) 3–5 | B2.1 improve one's general knowledge of locomotor system morphodynamics | 3.90 (±0.662) 3–5 |
| B1.2 are totally useless for improving one's general knowledge of locomotor system morphodynamics | 1.53 (±0.629) 1–3 | B2.2 are totally useless for improving one's general knowledge of locomotor system morphodynamics | 1.97 (±0.669) 1–3 |
| B1.3 are dynamic and interesting | 4.33 (±0.711) 3–5 | B2.3 are dynamic and interesting | 3.23 (±1.135) 1–5 |
| B1.4 are hard and boring | 1.60 (±0.675) 1–3 | B1.4 are hard and boring | 2.37 (±1.066) 1–4 |

[a] Likert scale: 1 = strongly disagree, 2 = disagree, 3 = neither agree nor disagree, 4 = agree, 5 = strongly agree; Cronbach's alpha = 0.876.

As shown in Table 3, responses to statement B1.1 (opposite B1.2) showed that a very high number of students stated that lessons, including clinical cases related to orthopedics and neurosurgery, improve general knowledge of locomotor system morphodynamics. Even answers to statement B2.1 (opposite B2.2) showed that a high number of students stated that lessons incorporating modeling computational biomechanical analysis principles improve one's general knowledge of locomotor system morphodynamics.

In Table 4, the results of the paired *t*-test with comparisons between statements of section B1 and section B2 are presented. Data revealed that student satisfaction was higher for the lessons that included clinical cases compared to satisfaction for the lessons that included computational biomechanical analysis principles.

**Table 4.** Paired *t*-test results with comparisons between statements of Sections B1 and B2.

| Statements | Mean Difference | T |
|---|---|---|
| B1.1 vs. B2.1 positive | 0.149 | 2.693 * |
| B1.2 vs. B2.2 negative | 0.141 | −3.067 ** |
| B1.3 vs. B2.3 positive | 0.216 | 5.086 *** |
| B1.4 vs. B2.4 negative | 0.207 | −3.699 *** |

The degrees of freedom are 29 for all tests. * $p \leq 0.05$; ** $p \leq 0.01$, *** $p \leq 0.001$.

Additionally, the answers to statement B1.3 (opposite B1.4) revealed that a very high number of students stated that the lessons, including clinical cases, were dynamic and interesting. A positive student lesson evaluation of the lessons that included modeling computational biomechanical analysis principles was also reflected by statement B2.3, even if this evaluation was lower than the evaluation of the surgical presentation.

However, the responses to the questions about the perceived low usefulness or boringness of these lectures show very low values, although, in this case, there is a statistical difference in the difficulty of the engineering lectures, which prevails over the difficulty found in the clinical integration lectures. There are statistically significant differences regarding perceived low usefulness for the clinical or engineering classes. Both show very low values but are significantly different from each other.

The Cronbach's alpha index value (0.876) revealed a good internal consistency for Sections A and B of the questionnaire.

### 3.2. Assessment of Student Knowledge of Locomotor System Anatomy

As shown in Table 5, results obtained in the assessment test of students' knowledge of the functional anatomy of the locomotor system showed differences between the two groups of students in the three sections of the test.

**Table 5.** Results obtained in the assessment test of students' knowledge of the functional anatomy of the locomotor system (Ns = number of students; Nq = number of questions).

| Test | Students | Ns | Nq | Mean (±SD) | 95% CI | Min | Max | ANOVA |
|---|---|---|---|---|---|---|---|---|
| Head and Trunk | HT | 30 | 11 | 7.0667 (±2.0833) | 6.2887–7.8446 | 2 | 11 | F = 7.911 ** |
| | Control | 32 | 11 | 5.4688 (±2.3689) | 4.6147–6.3228 | 1 | 10 | |
| Upper Limb | HT | 30 | 11 | 5.1000 (±1.5833) | 4.5088–5.6912 | 0 | 8 | F = 3.834 |
| | Control | 32 | 11 | 4.2188 (±1.9299) | 3.5229–4.9146 | 0 | 8 | |
| Lower Limb | HT | 30 | 8 | 2.5000 (±1.3582) | 1.9928–3.0072 | 0 | 5 | F = 9.454 ** |
| | Control | 32 | 8 | 3.7188 (±1.7271) | 3.0961–4.3414 | 0 | 7 | |

** $p \leq 0.01$.

In fact, in the 11 questions related to the functions of the head and trunk muscles, students in the HT course obtained significantly better results than students in the control course in terms of the number of correct answers.

In the 11 questions related to upper limb muscle functions, HT students also obtained better results, bordering on statistical significance, than control group students in terms of the number of correct answers (note that the significance level is only very slightly above the significance level of 0.05).

In contrast, the opposite result was obtained in the responses to eight questions related to lower limb muscle function. In fact, in this case, there was a significant prevalence of correct answers in the control medicine and surgery students compared to the correct answers given to the HT medicine and surgery students.

### 3.3. Correlation Analysis between Students' Views and Anatomy Knowledge Results

As shown in Table 6, there is a significant correlation between the evaluation of the usefulness of the two different types of clinical and engineering integration. Instead, these two values did not show any significant correlation with the three subtests measuring knowledge of the topic thought. A significant positive correlation emerged between the "head and neck" and "upper limb" subtests.

**Table 6.** Pearson correlation coefficients between the usefulness of surgical integration or bioengineering integration and the results obtained in the assessment test of students' functional anatomy knowledge (head and trunk, upper limb, lower limb).

| | Surgical Integration | Bioengineering Integration | Head and Trunk | Upper Limb |
|---|---|---|---|---|
| Bioengeneering Integration | 0.491 ** | | | |
| Head and Trunk | −0.197 | −0.168 | | |
| Upper Limb | −0.980 | 0.038 | 0.408 ** | |
| Lower Limb | −0.039 | 0.302 | 0.171 | 0.032 |

N = 30. ** Correlation significance ≤ 0.01 (two-tails).

## 4. Discussion

### 4.1. Curricular Integration in Medical Schools

The educational curricula of Italian medical schools are characterized by the presence of horizontal and vertical integration through the use of different interactive and multidisciplinary pedagogical approaches [5,7,9,15]. These innovative curricula can be

represented by a Z shape or as an inverted triangle structure, where students are introduced to clinical sciences at the beginning of the curriculum [5,7,9,15]. From a broader perspective, Italian curricula are now adapting to a general definition of vertical integration, allowing for a gradual involvement of students and young physicians in the professional community. This happens through a gradual increase in responsibilities in patient care, crossing the boundaries among undergraduate, postgraduate, and continuing medical education [5,16,17]. In the new Medicine and Surgery HT courses in international universities [1–3] and Italy, at Sapienza University of Rome [6], the curriculum is fully embedded with the disciplinary scientific field of biomedical engineering. This new curriculum needs new forms of integration with the human anatomy course, involving, for the first time, biomedical engineering.

*4.2. New Curricular Integration in the Human Anatomy Course: Students' Views*

In this general context, the surgical integration of human anatomy lectures, which usually occur in the first and second years of the degree program, as well as the presence of in-depth human anatomy lectures in later years within integrated clinical courses, represent long-recommended goals for improving human anatomy teaching and learning, contextualizing its learning concerning the procedures and clinicaßl skills to be acquired by the student [9,10,18–22]. Therefore, the clinical integration of anatomy is not new but has been established for many years in medicine and surgery Master's degree courses.

Anatomical knowledge is the basis of clinical procedures and the focus of the anatomy program should be centered on being propaedeutic to the clinical sciences [19]. The current trend is also to increase clinical integration in the human anatomy course to provide students with a stronger motivation to learn and understand every single topic [9,23,24]. In addition, this integration with technology provides students with a guiding principle for the problem-solving process that underlies the reasoning of a physician with technological expertise. The expertise that must underlie the problem-solving of this technological physician is precise and involves an integrated analysis of anatomy, physiology, pathophysiology, and technology [1]. The aim of integrating biomechanic skills into the human anatomy of the locomotor system course is to help students understand and apply robotic technologies related to joint surgery or spinal surgery, robotic surgery being an increasingly used technique in current surgical practice [25–29].

Our data showed that students like in-depth surgical integration much more than in-depth engineering integration. However, those who like in-depth surgical studies also like in-depth engineering studies. There was no significant correlation between the three test scores and levels of liking, while there was a significant correlation between students who liked clinical integration and those who liked engineering integration. A statistically significant correlation was also found in students who correctly answered questions on the head and trunk, with students answering correctly to questions on the upper limb.

The international literature lacks reports on this type of transdisciplinary teaching integration; our data are the first from an Italian HT Master's degree program in medicine and surgery and provide interesting insights. The greater preference for clinical integration vs. engineering integration is not surprising. The lectures given by the orthopedic surgeon and neurosurgeon showed clinical cases easily understandable even by the first-year students. In the surgically-integrated lectures, the anatomical aspects were emphasized, creating a strong link between surgery and anatomy. This had two effects: on the one hand, the students were motivated; on the other, they were allowed to contextualize the complex topics of the locomotor system anatomy. This was important to increase the otherwise misperceived usefulness of some topics in the anatomy program. Engineering-integrated lectures were also appreciated, even if to a lesser extent, being perceived as more complex lectures of the first year; this was due to a deeper knowledge of mathematics and calculus required to fully understand the usefulness of this type of lecture. Based on the results obtained in this study, the engineering-integrated lectures will be calibrated in the next

year of the course to delve into aspects of biomechanics at a level more appropriate for medical students.

### 4.3. New Curricular Integration in the Human Anatomy Course: Students' Knowledge of Functional Anatomy

Data obtained in the test of knowledge of the functional anatomy of the locomotor system showed significant differences in the higher number of correct answers given by the students in the HT course, compared with the students in the control group, in the head and trunk and upper limb sections of the test, while students in the control group significantly better answered the lower limb questions. The administered test contained no specific questions on the engineering topics covered in class, partly because the same test was administered to students from the other medical and surgical courses at Sapienza, where this type of integration was not present, and the questions were standard questions taken from a text used by all Italian medical students [14]. The addition of questions more relevant to engineering topics would probably have resulted in a better learning outcome but introduced a bias in the analysis on the actual usefulness of this type of integration.

The difference in results found in the different sections (or sub-tests) of the test may be because, in the HT medicine course, four fewer hours of lecture on the lower limb were provided (12 percent less than the total hours), compared to the hours of the lecture provided to the students in the control group, who thus had the opportunity to study this part of the program in greater depth. Another possible cause of these differences may be due to the personal preference in studying a specific topic, but unfortunately, preference for a specific topic was not asked about, so this could be an object of new studies. Further analysis on this point will help to specifically define the actual usefulness of engineering integration within the anatomy of the locomotor system course in the HT Master's degree program in medicine and surgery.

### 4.4. Study Limitations and Strengths

The limited number of students who participated in the study represents a limitation of the research, even though this new degree program has fewer students enrolled, and almost all of them have been involved in the study itself. Further analysis is needed to explain the differences in locomotor system anatomy test knowledge between the two groups of students analyzed.

### 4.5. Conclusions

This study, and the other in-depth analyses that will follow, are important in recalibrating and optimizing new modalities for curricular integration in human anatomy courses of the new Master's degree courses of Medicine and Surgery HT recently activated in Italy for the training of physicians with technological skills.

**Author Contributions:** Conceptualization, S.M., G.F. and P.F.; methodology, G.F., S.B., C.B., S.M. and P.F.; software, C.B.; validation, G.F., S.B., S.M., M.R. and P.F.; formal analysis, S.M., P.F. and M.R.; investigation, P.F., S.M., F.B., G.B. and C.B.; resources, M.R.; data curation, S.M. and P.F.; writing—original draft preparation, S.M., G.F. and P.F.; writing—review and editing, C.B., F.B. and S.B.; visualization, S.M. and P.F.; supervision, M.R., S.M. and P.F.; project administration, S.M. and P.F.; funding acquisition, S.M. All authors have read and agreed to the published version of the manuscript.

**Funding:** This research was funded by Sapienza University of Rome Ateneo 2021 grant number B83C22000960005.

**Institutional Review Board Statement:** Ethical review and approval were waived for this study due to its observance of the Italian University System's evaluation regulations administered by the National Agency for the Evaluation of the University System and Research (ANVUR-https://www.anvur.it/, accesed on 28 November 2022) and due to the observance of Sapienza University rules for anonymous students' satisfaction evaluation and Teaching Quality Assessment Questionnaire.

**Informed Consent Statement:** Informed consent was obtained from all subjects involved in the study.

**Conflicts of Interest:** The authors declare no conflict of interest.

## Appendix A

**Table A1.** Anatomy test questions. Locomotor system Anatomy—Functional Anatomy.

| | |
|---|---|
| The interspinal muscles' actions: | (a) flex the cervical spine<br>(b) extend the spine<br>(c) head on the neck rotation<br>(d) flex the cervical and lumbar spine<br>(e) contributes to respiratory movements |
| External oblique muscle action: | (a) bilaterally lowers the ribs and increases abdominal pressure<br>(b) unilaterally flexes and rotates the trunk homolaterally<br>(c) unilaterally rotates the thorax on the same side<br>(d) bilaterally raises the ribs and increases abdominal pressure<br>(e) bilaterally rotates the thorax on the opposite side |
| The coracobrachialis muscle: | (a) rotates the shoulder joint medially<br>(b) extends the shoulder joint<br>(c) adducts the shoulder joint<br>(d) adducts the shoulder joint<br>(e) laterally rotates the shoulder joint |
| Which of these statements is correct? | (a) the deltoid muscle is the main flexor of the arm<br>(b) the deltoid muscle is the main adductor of the arm<br>(c) the deltoid muscle is the main lateral rotator of the arm<br>(d) the deltoid muscle is the main medial rotator of the arm<br>(e) the deltoid muscle is the main abductor of the arm. |
| These statements are all true for the muscles of the anterior ligament of the leg except for one. Which one? | (a) they are responsible for the extension of the toes<br>(b) they are responsible for the dorsal flexion of the ankle<br>(c) they are innervated by the deep peroneal nerve<br>(d) they are innervated by the femoral and saphenous nerves<br>(e) they are responsible for the inversion/eversion of the ankle |
| Which of these statements is incorrect? | (a) the quadratus femoris muscle is considered an adductor muscle<br>(b) the gracilis muscle is considered an adductor muscle<br>(c) the pectineus muscle is considered an adductor muscle<br>(d) the long adductor muscle is considered an adductor muscle<br>(e) the short adductor muscle is considered an adductor muscle |
| Which of these statements is incorrect? | (a) the coracobrachialis muscle adducts the shoulder joint<br>(b) the coracobrachialis muscle flexes the shoulder joint<br>(c) the great round muscle rotates the arm medially<br>(d) the deltoid muscle flexes the arm<br>(e) the infraspinatus muscle rotates the arm medially |
| The semimembranosus muscle is | (a) lateral rotator of the thigh<br>(b) flexor of the leg and extensor of the thigh<br>(c) thigh flexor<br>(d) extensors of the leg<br>(e) adductor of the thigh |
| Which of these statements is incorrect? | (a) the great round muscle flexes the arm<br>(b) the coracobrachialis muscle flexes the shoulder joint<br>(c) the coracobrachialis muscle adducts the shoulder joint<br>(d) the infraspinatus muscle rotates the arm medially<br>(e) the great round muscle rotates the arm medially |

**Table A1.** *Cont.*

| | |
|---|---|
| Which of these muscles are the external rotators of the hip? | (a) the external obturator muscle<br>(b) the gluteus medius muscle<br>(c) the great gluteus muscle<br>(d) the gluteus minimus muscle<br>(e) the tensor fascia lata muscle |
| Which of these muscles are the internal rotators of the hip? | (a) the great gluteus muscle<br>(b) the quadratus femoris muscle<br>(c) the internal obturator muscle<br>(d) the gluteus medius muscle<br>(e) the external obturator muscle |
| The long head of the triceps brachii muscle is a: | (a) supinator of the forearm<br>(b) extensor of the elbow<br>(c) flexor of the elbow<br>(d) pronator of the forearm<br>(e) abductor of the shoulder |
| Which of these statements is correct? | (a) the teres minor muscle rotates the arm laterally.<br>(b) the teres minor muscle adducts the arm<br>(c) the teres minor muscle adducts the arm<br>(d) the teres minor muscle flexes the arm<br>(e) the teres minor muscle rotates the arm medially |
| The brachialis muscle is a: | (a) supinator of the hand<br>(b) elbow flexor<br>(c) pronator of the hand<br>(d) extensor of the elbow<br>(e) wrist abductor |
| Bilateral contraction of the splenius cervicis muscles: | (a) flexes the neck<br>(b) tilts the neck laterally<br>(c) tilts the neck laterally from the opposite side<br>(d) extends the neck<br>(e) rotates the neck |
| The action of the medial rectus muscle results in the following: | (a) abduction of the eye<br>(b) adduction of the eyes<br>(c) looking up and medially at<br>(d) looking down and laterally<br>(e) looking down and medially |
| The rectus abdominis muscle: | (a) flexes the trunk forward<br>(b) with a fixed point on the trunk, acting individually, rotates the pelvis<br>(c) with a fixed point on the trunk flexes the lower limb on the pelvis<br>(d) acting individually, flexes the trunk laterally<br>(e) contributes to inhalation |
| The coccygeal muscle determines | (a) elevation and support of the pelvic floor<br>(b) extension of the coccygeal joints<br>(c) retroposition of the anal canal<br>(d) depression of the pelvic floor<br>(e) nutation of the sacrum |
| Which of these statements is correct? | (a) the gastrocnemius muscle extends the leg<br>(b) the gastrocnemius muscle is an extensor muscle<br>(c) the gastrocnemius muscle extends the ankle<br>(d) the gastrocnemius muscle is one of the extrinsic muscles of the foot<br>(e) the gastrocnemius muscle is one of the intrinsic muscles of the foot |

**Table A1.** *Cont.*

| | |
|---|---|
| The levator muscle of the upper eyelid determines | (a) closing of the eyes<br>(b) lifting of the eyebrow<br>(c) elevation of the lower eyelid<br>(d) elevation of the upper eyelid<br>(e) wrinkling of the forehead |
| Which of these statements is correct? | (a) the iliopsoas muscle flexes the lumbar spine<br>(b) the iliopsoas muscle adducts the lumbar spine<br>(c) the iliopsoas muscle extrude the lumbar spine<br>(d) the iliopsoas muscle adducts the lumbar spine<br>(e) the iliopsoas muscle is an internal rotator of the lumbar spine |
| Which of these statements is correct? | (a) the piriformis muscle adducts the lumbar spine<br>(b) the psoas major muscle flexes the lumbar spine<br>(c) the iliopsoas muscle intrudes the lumbar spine<br>(d) the long adductor muscle extends the lumbar spine<br>(e) the iliopsoas muscle adducts the lumbar spine |
| Which of these muscles extends the femur | (a) the gluteus minimus muscle<br>(b) the tensor fascia lata muscle<br>(c) the piriformis muscle<br>(d) the gluteus maximus muscle<br>(e) the gluteus medius muscle |
| Which of these statements is correct? | (a) the subscapularis muscle adducts the arm<br>(b) the subscapularis muscle flexes the arm<br>(c) the subscapularis muscle rotates the arm laterally<br>(d) the subscapularis muscle adducts the arm<br>(e) the subscapularis muscle rotates the arm medially |
| The medial head of the triceps brachii muscle is a: | (a) supinator of the forearm<br>(b) extensor of the elbow<br>(c) elbow flexor<br>(d) pronator of the forearm<br>(e) abductor of the shoulder |
| Which of these statements is incorrect? | (a) the coracobrachialis muscle flexes the shoulder joint<br>(b) the coracobrachialis muscle adducts the shoulder joint<br>(c) the teres major rotates the arm medially<br>(d) the subscapularis muscle rotates the arm medially<br>(e) the coracobrachialis muscle adducts the shoulder joint |
| The action of the palatine veil elevator and palatine veil tensor muscles is to: | (a) elevates the soft palate<br>(b) constrict the pharynx<br>(c) elevate the pharynx<br>(d) constrict the isthmus of the jaws<br>(e) elevate the larynx |
| The piriform muscle: | (a) extends the femur<br>(b) intrarotates the hip<br>(c) adducts the hip<br>(d) flexes the lumbar spine<br>(e) flexes the femur |
| The extensor digitorum communis muscle: | (a) flexes all fingers<br>(b) has a common origin with the extensor carpi radialis brevis muscle<br>(c) flexes the wrist<br>(d) flexes the thumb<br>(e) extends the thumb |
| The external anal sphincter muscle: | (a) raises the anal canal<br>(b) narrows and lowers the anal canal<br>(c) closes the anal canal and the anus<br>(d) narrows and lifts the anal canal<br>(e) provides involuntary control for defecation |

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
