# Peer review of "Surgical and Bioengineering Integration in the Anatomy Course of Medicine and Surgery High Technology: Knowledge and Perception of Anatomy"

_2813-0545, doi:10.3390/anatomia2010006_

Round 1
Reviewer 1 Report
The study could be interesting for another fields, maybe related to social o teach media, but I don´t think it will be interesting for a health sciences and medical research
Author Response
Question Reviewer 1
The study could be interesting for another fields, maybe related to social o teach media, but I don´t think it will be interesting for a health sciences and medical research
Answer from the Authors
Dear Reviwer, our study does not focus on the use of social media or particular software or devices for teaching, but intends to evaluate the delivery of in-person lectures in which the anatomy curriculum has been integrated with clinical and bioengineering content. therefore, it is an evaluation on the integration of the curriculum and the lectures are conducted in the normal manner of in-person lectures so in our opinion the topic of the article is fully within the scope of the journal Anatomy.
Reviewer 2 Report
The objective and scope of the manuscript Ms. No. 2097114 could be of interest to Anatomia but needs major changes and full re-review.
MM: The consistency of the questionnaires (Cronbach's alpha index) should be indicated in this section
L154: The paper referenced here is a study in neuroanatomy but not in the locomotor system. This reference must be changed or otherwise indicate the consistency of the questionnaire (Cronbach's alpha index) used.
Results: The text of this section should not duplicate the information contained in the tables, only highlight what is relevant or significant.
Tables: indicate significant values, for example with asterisks as in Table 6
Discussion: to facilitate the reading of this section, indicate in independent sections: 1) the limitations and strengths of the study and 2) conclusion
Despite of addressing an important aspect, I consider this manuscript not to be acceptable in its present form for publication in Anatomia
Author Response
MM: The consistency of the questionnaires (Cronbach's alpha index) should be indicated in this section
Answer:The Cronbach's alpha index has been indecated in the section as requested
L154: The paper referenced here is a study in neuroanatomy but not in the locomotor system. This reference must be changed or otherwise indicate the consistency of the questionnaire (Cronbach's alpha index) used.
Answer: Even in this case the Cronbach's alpha index has been indecated in the section as requested.
Results: The text of this section should not duplicate the information contained in the tables, only highlight what is relevant or significant.
Answer: All the duplicate information have been removed from the text as requested.
Tables: indicate significant values, for example with asterisks as in Table 6
Answer: Significant values have been indicated by asteriskd as requested.
Discussion: to facilitate the reading of this section, indicate in independent sections: 1) the limitations and strengths of the study and 2) conclusion
Answer: the discussion was implemented and divided in sections as requested.
Reviewer 3 Report
These authors discuss on how education in locomotor system anatomy with neurosurgery, orthopedic and 2 bioengineering integrations takes place in master's degree course medicine and surgery HT at Sapienza University of Rome. They analyze student’s knowledge and point of view in this regard.
Vertical integration of basic and clinical sciences in medical curriculum is an idea that is presemt for a long time. I recommend (if I may) checking Rosenthal DR, Worley PS, Mugford B, Stagg P. Vertical integration of medical education: Riverland experience, South Australia. Rural Remote Health. 2004 Jan-Mar;4(1):228. Epub 2004 Jan 8. PMID: 15882104. on their experience. There you can compare yours point of view with their definition of vertical integration and see how you are in line with grouping of curricular content and delivery mechanisms. You should also benefit of Wijnen-Meijer M, van den Broek S, Koens F, Ten Cate O. Vertical integration in medical education: the broader perspective. BMC Med Educ. 2020 Dec 14;20(1):509. doi: 10.1186/s12909-020-02433-6. PMID: 33317495; PMCID: PMC7737281.
To start with title, it needs to be rewritten – short and informative. The rest of manuscript is fine.
Author Response
We thank the reviewer for the important suggestions, useful to improve the manuscript. We add the references proposed in the text of the manuscript and we discussed them in the text.
Round 2
Reviewer 1 Report
I think now is a paper to publish
Reviewer 2 Report
Congratulations!!
Reviewer 3 Report
much improved